# Structural Characterization and Assessment of Anti-Inflammatory Activities of Polyphenols and Depsidone Derivatives from *Melastoma malabathricum subsp. normale*

**DOI:** 10.3390/molecules27051521

**Published:** 2022-02-24

**Authors:** Rui-Jie He, Ya-Feng Wang, Bing-Yuan Yang, Zhang-Bin Liu, Dian-Peng Li, Bi-Qun Zou, Yong-Lin Huang

**Affiliations:** 1Guangxi Key Laboratory of Plant Functional Phytochemicals and Sustainable Utilization, Guangxi Institute of Botany, Guangxi Zhuang Autonomous Region and Chinese Academy of Sciences, Guilin 541006, China; hrj937@gxib.cn (R.-J.H.); wyf@gxib.cn (Y.-F.W.); yby@gxib.cn (B.-Y.Y.); lzb@gxib.cn (Z.-B.L.); ldp@gxib.cn (D.-P.L.); 2Department of Chemistry, Guilin Normal College, Gulin 541199, China

**Keywords:** *Melastoma malabathricum subsp. normale* (D. Don) Karst. Mey, *Melastoma*, polyphenols, complex tannin, depsidone derivatives, anti-inflammatory

## Abstract

The roots of *Melastoma malabathricum subsp. normale* (D. Don) Karst. Mey have been used in traditional ethnic medicine systems in China to treat inflammation-triggered ailments, such as trauma, toothache, and fever. Therefore, the aim of this study is to screen for compounds with anti-inflammatory activity in the title plant. The extract of *M*. *malabathricum subsp. normale* roots was separated using various chromatographic methods, such as silica gel, ODS C18, MCI gel, and Sephadex LH-20 column chromatography, as well as semi-preparative HPLC. One new complex tannin, named whiskey tannin D (**1**), and an undescribed tetracyclic depsidone derivative, named guanxidone B (**2**), along with nine known polyphenols (**2**–**10**) and three known depsidone derivatives (**12**–**14**) were obtained from this plant. The structures of all compounds were elucidated by extensive NMR and CD experiments in conjunction with HR-ESI-MS data. All these compounds were isolated from this plant for the first time. Moreover, compounds **1**–**4**, **8**, and **10**–**14** were obtained for the first time from the genus *Melastoma*, and compounds **1**, **2**, and **11**–**14** have not been reported from the family Melastomataceae. This is the first report of complex tannin and depsidone derivatives from *M. malabathricum subsp. normale*, indicating their chemotaxonomic significance to this plant. Compounds **1**–**12** were investigated for their anti-inflammatory activities on the production of the nitric oxide (NO) in lipopolysaccharide (LPS)-stimulated RAW264.7 cells, and compounds **1**, **11**, and **12** showed anti-inflammatory activities with IC_50_ values of 6.46 ± 0.23 µM, 8.02 ± 0.35 µM, and 9.82 ± 0.43 µM, respectively. The structure–activity relationship showed that the catechin at glucose C-1 in ellagitannin was the key to its anti-inflammatory activity, while CH_3_O- at C-16 of aromatic ring A in depsidone derivatives had little effect on its anti-inflammatory activity. The study of structure–activity relationships is helpful to quickly discover new anti-inflammatory drugs. The successful isolation and structure identification of these compounds, especially complex tannin **1**, not only provide materials for the screening of anti-inflammatory compounds, but also provide a basis for the study of chemical taxonomy of the genus *Melastoma*.

## 1. Introduction

The genus *Melastoma* (Melastomataceae), with approximately 100 species, is widespread in southern Asia, northern Oceania, and the Pacific islands, and a total of 9 species and 1 variety are found in China [1]. Some species of this genus are used for the treatment of diarrhea, dysentery, leucorrhoea, ulcers, and wounds [2]. Flavonoids, tannins, phenylpropanoids, organic acids (esters), terpenoids, and other components were previously characterized from this genus [3]. Some of them exhibited anti-inflammatory [4], hemostatic activity [5], anticoagulant activity [6], antibacterial activity [7], antioxidant activity [8,9], hepatoprotective activity [10], gastroprotective activity [11], hypoglycemic activity [12], and cytotoxic activities [13]. *Melastoma malabathricum subsp. normale* (D. Don) Karst.Mey, a shrub of the family Melastomataceae, grows mainly in Xizang, Sichuan, Guangxi, and Fujian provinces of China [1]. Its roots have been used in Zhuang and Yao medicines for the treatment of inflammation-triggered ailments, such as trauma, toothache, and fever [14,15]. With the aim to find compounds with anti-inflammatory activity in the title plant*,* the roots of *M. malabathricum subsp. normale* were extracted by 80% aqueous acetone, and subsequently separated using silica gel, MCI, ODS C18, and Sephadex LH-20 column chromatography, as well as semi-preparative HPLC to yield ten polyphenols and four depsidone derivatives. The structures of these compounds were characterized by experimental and published spectroscopic data analyses. As we all know, complex tannin is a kind of flavono-ellagitannin, which has a unique C-C condensation structure of C-glycoside tannin (vescalagin-type or stachyurin-type) and flavane-3-alcohol. To date, these compounds have only been found in a few plant families, including Combretaceae, Myrtaceae, Melastomataceae, Fagaceae, and Theaceae [16]. Compound **1** is the only complex tannin isolated from *M. malabathricum subsp. normale*, and its analogs have also been reported from *M*. *malabathricum* L. in the family Melastomataceae [17], suggesting their closely chemotaxonomic relationships between *M. malabathricum subsp. normale.* and *M*. *malabathricum*. L. Depsidone derivatives have never been reported from the family Melastomataceae [3,18]. These compounds enrich the chemical diversity of *M. malabathricum subsp. normale* and provided a basis for the chemotaxonomic studies of the species of the genus *Melastoma*. Moreover, the anti-inflammatory activities of compounds **1**–**12** were investigated to develop polyphenols or depsidone derivatives as a novel anti-inflammatory drug. In the present study, the isolation and structural elucidation of compounds **1** and **11**, as well as the anti-inflammatory activities of **1**–**12**, are reported in detail.

## 2. Results and Discussion

The EtOAc fractionation and purification of the 80% aqueous acetone extract of *M. malabathricum subsp. normale* roots using various chromatographic methods yielded ten polyphenols (**1**–**10**) and four depsidone derivatives (**11**–**14**). The known compounds **2**–**10** and **12**–**14** were identified by analysis of mass spectral data, the NMR spectral data, specific rotations, and/or melting point data as whiskey tannin B (**2**) [19]; castalagin (**3**) [20]; 3,3′-dimethoxy ellagic acid (**4**) [21]; 3,3′,4-trimethoxyellagic acid (**5**) [22]; 3,3′,4′-trimethoxyellagic acid-4-*O*-β-d-glucopyranoside (**6**) [23]; 3,3′-dimethoxy ellagic acid-4-*O*-α-d-xylopyranoside (**7**) [24]; 1,2,4-benzenetriol (**8**) [25]; 1,4,6-tri-*O*-galloyl-glucose (**9**) [26]; 6-*O*-galloyl-glucose (**10**) [27]; guanxidone A (**12**) [28]; excelsione (**13**) [29]; and dioxepin-11-one (**14**) [30]. The structures of **1**–**1****4** are shown in Figure 1. All these compounds were obtained from this species for the first time. Moreover, compounds **1**–**4**, **8**, and **10**–**14** were isolated for the first time from the genus *Melastoma*, and compounds **1**, **2**, and **11**–**14** were reported from the family Melastomataceae for the first time.

### 2.1. Structure Elucidation

Compound **1**, a pale brown amorphous powder, shows the positive coloration characteristic of complex tannin when reacting with anisaldehyde-sulfuric acid (pink) and NaNO_2_-AcOH (brown) reagent. A deprotonated molecular ion peak at *m*/*z* 1249.1580 [M − H]^−^ (calcd, 1249.1586) was observed in the HR-ESI-MS spectrum, indicating that the molecular formula of **1** is C_58_H_42_O_32_. The ^1^H NMR data (Table 1) revealed at least two hexahydroxydiphenoyl (HHDP) groups at δ 6.63 (s), 6.80 (s), and 7.02 (s); an ethoxyl at δ 4.21 (q, 7.1, 2H) and 1.20 (t, 7.1, 3H). As shown in Figure 1, the ^1^H-^1^H COSY correlations among methylene (δ 3.86 and 4.81) and five methine protons (δ 4.36–5.53) revealed a polyalchohol unit, which exhibited similar NMR data to the open-chain glucose core of stenophyllanine B [31]. The ^13^C NMR data (Table 1) revealed six ester carbonyl groups at δ 170.3, 168.2, 168.0, 167.8, 167.4, and 164.2. Five downfield signals at δ_C_ 63.6–83.2 suggested that the hydroxyl at C-2–C-6 was esterified. The large difference in chemical shifts between H-6a (δ 4.81) and 6b (δ 3.86) suggested one of the HHDP moieties was located at C-4 and C-6, which can be explained by the anisotropic effect of a C-6 ester carbonyl group. It is restrained to be rigidly coplanar with one of the C-6 methylene protons in the eleven-membered diester ring, so the proton was placed in a strongly deshielding environment [32]. This was also supported by the correlations of H-4 and H-6 with carbonyl carbons C_HHDP-7′′′__′_ (δ 168.2) of the HHDP group and C_HHDP-7′′′′′_ (δ 167.4) of the HHDP group in the HMBC spectrum (Figure 2). The carbon signals at δ 201.4 (Cp-3′), 170.3 (Cp-7′), 155.9 (Cp-5′), 146.8 (Cp-4′), 84.1 (Cp-2′), 45.1 (Cp-1′), 62.9 (OCH_2_), and 14.3 (CH_3_) were assignable to a cyclopentenone ring bearing an ethoxycarbonyl moiety. This was confirmed by the correlations of methylene protons (δ 4.21) with Cp-7′ and H_Cp_-1′ (δ 5.60) with Cp-2′, Cp-3′, Cp-4′, Cp-5′, and Cp-7′ in the HMBC spectrum. In addition, the HMBC spectrum showed correlations of H-1 with Cp-3′, Cp-4′, and Cp-5′, indicating the linkages of C-1 with Cp-4′. The carbonyl signals at δ 164.2 (Cp-6′) suggested that this carbonyl was connected by a double bond and formed a δ-lactone ring with glucose *O*-2. This was confirmed by the correlation of H-2 with carbonyl carbon (δ 164.2) in the HMBC spectrum. The correlations of H-3 with C_HHDP-7″_ (δ 168.0) of the HHDP group and H-5 with C_HHDP-7′′′_ (δ 167.8) of the HHDP group in the HMBC spectrum indicated that these two carbonyl carbons (C_HHDP-7″_ and C_HHDP-7′′′_) were connected to glucose *O*-3 and glucose *O*-5, respectively. The correlations of H_Cp_-1′ with C_HHDP-2″_ (δ 124.7) and C_HHDP-3″_ (δ 112.8) of the HHDP group in the HMBC spectrum indicated that Cp-1′ was linked to C_HHDP-3_ of the HHDP group. Comparison between the NMR data of whiskey tannin B [19] and **1** (Table 1) revealed that -OH at C-1 in whiskey tannin B was replaced by a 5,7,3′,4′-tetrahydroxy flavan-3-ol moiety in **1**. This moiety could be constructed by analysis of the ^1^H NMR data of a 1,2,4-trisubstituted aromatic ring at δ 6.94 (d, 0.9), 6.86 (dd, 8.2, 0.9), and 6.78 (d, 8.2) and a phloroglucinol aromatic ring at δ 6.00 (s), as well as the C-ring characteristic protons of a 2,3-trans flavan-3-ol at 4.72 (br s), 4.11 (m), 2.57 (dd, 16.1, 7.7), and 2.83 (d, 16.1) [33]. This is further supported by the ^1^H-^1^H COSY correlations of catechin H-3′′′′′′ with catechin H-2′′′′′′ and catechin H-4′′′′′′ (Figure 1), and by the correlations of catechin H-2′′′′′′ with aromatic carbons (114.2 and 120.6) and catechin H-3′′′′′′ with aromatic carbon (δ 131.3) in the HMBC spectrum. This moiety has also been found in stenophyllanine B [31]. To determine the C-6 or C-8 linkage between catechin and C-glycosylated ellagitannin moieties, methylation of **1** was carried out, giving **1**a, and its ^13^C-NMR data showed an unsubstituted A-ring carbon signal at δ 89.6, indicating the presence of a substituent at C-8 of the flavan-3-ol moiety [34]. The side-chain moiety linked to C-1 was confirmed by the correlations of H-1 with aromatic carbons (δ 102.9, 154.9, and 156.2) in the HMBC spectrum and by the upfield chemical shift (δ 33.9). Compound **1** was refluxed in 20% acetic acid ethanol, and then chromatographed on Sephadex LH-20 to obtain a crystalline compound [m.p. 170–171 °C; ${\left[\alpha \right]}_{D}^{25}$ + 14° (acetone)] that was identical with (+)-catechin [34]. Thus, the planar of **1** was identified.

The coupling constant between H-1 and H-2 is 0 Hz (<2.0 Hz), indicating that the configuration at C-1 of the glucose core in **1** is the same as that of vescalagin (*J* = 2.0 Hz) [35] and different from that of whisky tannin B (*J* = 6.4 Hz) [19]. This is also evidenced by the nuclear Overhauser effect (NOE) correlations of H-1 with H-3. Assuming that **1** is derived from vescalagin, inspection of a Dreiding model of **1** showed that the proton Hc_p-1′_ of the cyclopentenone ring must be *β* oriented because its fusion ring system is so rigid that it is impossible to build an alternative model [35]. No proton was correlated with Hc_p-1′_ in the ROESY spectrum indicated that the ethoxycarbonyl in **1** is α-orientation. The 2*R-* and 3*S*- configurations of the flavan C-ring were deduced from the absence of the NOE cross peaks between catechin H-2′′′′′′ and catechin H-3′′′′′′ in the ROESY spectrum of **1**, as well as acid hydrolysis of **1** gave (+)-catechin. The atropisomerism of the HHDP group in **1** was determined to be *S*, as indicated by a positive Cotton effect at 240 nm and a negative one at 265 nm in its CD spectrum [36]. Thus, the structure of compound **1**, named whiskey tannin D, was characterized as depicted in Figure 1.

Compound **11** was obtained as a white powder and had a molecular formula of C_18_H_14_O_7_ based on HR-ESI-MS (*m*/*z* 341.0669 [M − H]^−^, calcd 341.0661) and NMR data (Table 2), requiring 12 degrees of unsaturation. MS analysis and evaluation of NMR data suggested that compound **1****1** was a tetracyclic depsidone [28]. The ^1^H NMR data of **1****1** showed one singlet at δ_H_ 6.63 (1H, s, H-8), one oxy-methylene singlet at δ_H_ 5.25 (2H, s, H-15), and three aromatic methyl singlets at δ_H_ 2.30 (6H, s, H-16, 17) and 2.12 (3H, s, H-18). The ^13^C NMR data exhibited 18 carbon signals, in addition to the three signals due to the methyl groups (δ_C_ 8.7, C-16; 11.0, C-18; and 20.4, C-17), and fourteen resonances attributable to a tetracyclic depsidone containing two carbonyl carbons at δ_C_ 168.9 and 161.7. The HMBC (Figure 3) of δ_H_ 6.63 (H-8) with δ_C_ 20.4 (C-17), 111.4 (C-10), and 114.9 (C-6), and of δ_H_ 2.30 (3H, s, H-16) with δ_C_ 160.4 (C-5), 160.5 (C-7), and C-6 were observed, indicating that **1****1** possessed a pent-substituted aromatic ring A. The HMBC from δ_H_ 5.25 (H-15) to δ_C_ 168.9 (C-1), 109.5 (C-2), 143.7 (C-14), 113.9 (C-13), and 148.6 (C-12) indicated that an oxymethylene is situated at position 14 of aromatic ring C. Furthermore, the HMBC from δ_H_ 2.12 (H-18) to δ_C_ 148.6 (C-12), 113.9 (C-13), 143.7 (C-14), and 140.1 (C-4) demonstrated that another methyl group is linked to position 13 of aromatic ring C. Detailed analysis of the 1D NMR data of **1****1** (Table 2) revealed high structural similarity to the co-isolated excelsione (**1****3**) [29]. The only difference between **1****1** and **1****3** was the replacement of the -CH_2_OH (C-16) by a methyl group, which was supported by the chemical shifts of C-16 (δ_C_ 8.7). The structure of **11** was therefore established and named guanxidone B.

### 2.2. Anti-Inflammatory Activity Assays

All compounds except compounds **13** and **14** were investigated for potential anti-inflammatory activity by measuring the inhibition of the nitric oxide (NO) production. As shown in Table 3, compounds **1**, **11**, and **12** displayed significant anti-inflammatory activity with IC_50_ values ranging from 6.46 ± 0.23 to 9.82 ± 0.43 μM. The IC_50_ values for the inhibition of NO production by other compounds are all > 10 μM. The anti-inflammatory activity of compound **1** is better than that of compound **2**, indicating that the effect of catechin on glucose C-1 is very important for its activity. Compound **11** has better anti-inflammatory activity than compound **12**, indicating that CH_3_O- at C-16 of aromatic ring A has little effect on its activity.

## 3. Experimental

### 3.1. Materials

The roots of *M*. *malabathricum subsp*. *normale* were collected in Yanshan Town (Guilin, China) in September 2019, and identified by Professor Yusong Huang (Guangxi Institute of Botany, Guilin, China). A voucher specimen (registration No. 20190915) has been deposited in the Guangxi Key Laboratory of Plant Functional Phytochemicals and Sustainable Utilization Guangxi Institute of Botany, Guilin, China.

### 3.2. General Experimental Procedures

Optical rotations were measured at 25 °C with an ADP440+ polarimeter, Julabo, Seelbach, Germany (λ 589 nm, path length 1.0 cm). The UV spectra were recorded in MeOH on a TU-1901 spectrophotometer (Beijing Puxi General Instrument Co., Ltd., Beijing, China). The CD spectra were acquired in MeOH on a JASCO J-180 spectropolarimeter (Jasco, Tokyo, Japan). The NMR spectra were obtained on a Brucker Avance III HD-500 MHz spectrometer (Bruker Biospin AG, Fällanden, Switzerland), and the residual solvent peaks were used as references. Coupling constants and chemical shifts were given in Hz and on a δ (ppm) scale, respectively. ESI-MS and HR-ESI-MS were acquired on a Bruker Esquire 3000plus and Waters/Micromass Q-TOF-Ultima (Waters, Milford, MA, USA) mass spectrometers, respectively. Semi-preparative HPLC performed on a Shimadzu LC-20AT HPLC system at the rate of 2 mL/min. Sephadex LH-20 (GE Healthcare Bio-Science AB, Uppsala, Sweden), MCI gel CHP 20P (Mitsubishi Chemical Co., Tokyo, Japan), silica gel (Qingdao Marine Chemical Co., Ltd., Qingdao, China), and Chromatorex ODS (Merck, Darmstadt, Germany) were used for column chromatography (CC).

### 3.3. Extraction and Separation

Air-dried, powdered roots of *M. malabathricum subsp. normale* (9.0 kg) were extracted with the 80% aqueous acetone for three times (each for 7 days) at room temperature to afford a residue (0.6 kg). Then, the residue was suspended in H_2_O (1 L) and successively partitioned with petroleum ether, EtOAc into petroleum ether (Fraction A, 175.0 g), EtOAc (Fraction B, 80.0 g), and water (Fraction C, 345.0 g) fractions. Fraction B (80.0 g) was chromatographed on silica gel column (10 × 30 cm) and eluted with a gradient of MeOH-CH_2_Cl_2_ (0:100–100:0, *v*/*v*) to afford ten fractions (Fr.B1–Fr.B10). Fr.B4 (15.0 g) was purified by ODS C18 CC (6 × 50 cm) eluting with a gradient of MeOH-H_2_O (0:100–100:0, *v*/*v*) to obtained eighteen subfractions (Fr.B4-1–Fr.B4-18). Further separation of Fr.B4-6 (5.6 g) using silica gel column (4 × 20 cm) in a gradient of MeOH-CH_2_Cl_2_ (10:100–40:60, *v*/*v*) and then Sephadex LH-20 CC (1.5 × 40 cm) in a gradient of MeOH-H_2_O (0:100–100:0, *v*/*v*, 10% stepwise, each 200 mL) to give compounds **2** (34.0 mg) and **3** (8.1 mg). Separation of Fr.B4-7 (7.2 g) was done by silica gel column (5 × 20 cm) eluting with a gradient of MeOH–CH_2_Cl_2_ (0:100–20:80, *v*/*v*) and then Sephadex LH-20 CC eluting with MeOH-CH_2_Cl_2_ (1:1, *v*/*v*) to yield **1** (28.2 mg), **8** (5.3 mg), **9** (14.3 mg), **11** (8.8 mg), and **12** (7.6 mg). Compounds **13** (1.1 mg) and **14** (1.3 mg) were obtained from Fr 4-12 (0.6 g) successively via semi-preparative HPLC eluted with a gradient of MeOH-H_2_O (50:50–90:10, *v*/*v*, 0–40 min) and Sephadex LH-20 CC eluted with CH_2_Cl_2_-MeOH (1:1, *v*/*v*).

Fraction C (345.0 g) was divided into twenty fractions (Fr.C1–Fr.C20) by Sephadex LH-20 CC (10 × 45 cm) eluting with a gradient of MeOH-H_2_O gradients (0:100–100:0, *v*/*v*, 20% stepwise, each 5000 mL). Fr.C1 (40.0 g) was subjected to HP20SS column (6 × 60 cm) with a gradient of MeOH-H_2_O (0:100–100:0, *v*/*v*) to give nine subfractions (Fr.C1-1–Fr.C1-9). Fr.C1-5 (12.0 g) was further chromatographed on ODS C18 column (4 × 30 cm) using MeOH-H_2_O step gradient (0:100–100:0, *v*/*v*), then purified by Sephadex LH-20 eluting with MeOH to yield compounds **4** (3.8 mg), **5** (2.2 mg), **6** (2.5 mg), and **7** (5.4 mg). Fr.C3 (10.0 g) was loaded onto a MCI gel CHP 20P (4 × 30 cm) column and eluted with a gradient of MeOH-H_2_O (0:100–100:0, *v*/*v*) to afford 10 fractions (Fr.C3-1–Fr.C3-10). Compound **10** (2.3 mg) was obtained from Fr.C3-3 (2.8 g) using Sephadex LH-20 column (2 × 40 cm) in a gradient of MeOH-H_2_O (0:100–50:50, *v*/*v*, 10% stepwise, each 300 mL).

### 3.4. Spectroscopic Data

Whiskey tannin D (**1**): A pale brown amorphous powder; ${\left[\alpha \right]}_{D}^{25}$ − 15.6° (*c* = 0.12, MeOH); UV (MeOH) λmax nm (log ε): 204 (2.12), 272 (1.20); CD (MeOH) λmax (Δε) 263 (−6.6), 240 (+9.5) (Appendix A). ^1^H and ^13^C NMR data, see Table 1; 1D and 2D NMR spectra of 1 (Appendix A); HR-ESI-MS *m*/*z*: 1249.1580 [M − H]^−^ (calcd, 1249.1586) (Appendix A).

Guanxidone B (**11**): a white powder; UV (MeOH) λ_max_ nm (log *ε*): 206 (1.60). ^1^H and ^13^CNMR data, see Table 2; 1D and 2D NMR spectra of 11 (Appendix A, Appendix A); HR-ESI-MS *m*/*z*: 341.0669 [M − H]^−^ (calcd, 341.0661) (Appendix A, Appendix A).

### 3.5. Acid-Catalyzed Degradation of ***1***

Compound **1** (10 mg) was dissolved in 20% acetic acid ethanol (3 mL) and reacted under reflux for 5 d. The solvent was concentrated under reduced pressure, and the residue was chromatographed on Sephadex LH-20 column eluting with ethanol to yield a colorless needles (+)-catechin (3.5 mg) [m.p. 170–171 °C; ${\left[\alpha \right]}_{D}^{25}$ + 14° (acetone)].

### 3.6. Methylation of ***1***

Compound **1** (10 mg) was dissolved in dimethyl sulfate (1 mL), then anhydrous potassium carbonate (0.5 g) in acetone was added and heated under reflux for 3 h. After removing inorganic salts by filtration, the filtrate was evaporated off under reduced pressure, and loaded onto a silica gel CC gradually eluting with benzene with increased proportion of acetone to yield compound **1a** (2.1 mg), ${\left[\alpha \right]}_{D}^{25}$ − 123.0° (c = 0.42, CHCl_3_). HR-ESI-MS *m*/*z*: 1473.1095 [M − H]^−^ (Calcd. 1473.1090 for C_74_H_73_O_32_^−^). The flavan part of **1**a: ^1^H-NMR (CDCl_3_) δ 5.71 (1H, d, *J* = 8.9 Hz, 5”-H), 6.27 (1H, s, 6-H), 6.61–6.93 (6H, m, aromatic H). ^13^C-NMR (CDCl_3_): δ 27.8 (C-4), 38.1 (C-1”), 65.7 (C-6”), 68.1 (C-3), 69.5, 70.2, 71.5, 76.5 (C-2”, C-3”, C-4” and C-5”), 82.5 (C-2), 89.6 (C-6) [34].

### 3.7. Anti-Inflammatory Activity

The anti-inflammatory activities of compounds **1**–**12** were investigated on the production of the NO in LPS-stimulated cells according to our previously described method [17]. That is, the RAW 264.7 cells were cultivated in DMEM supplemented with 10% FBS at 37 °C for 24 h. Cells in 24-well plate were treated with 200 ng/mL LPS and the test compounds. After 22 h, the media were collected, and the level of nitrite was measured using the Griess Reagent reagent System (Promega), Madison, WI, USA. The results are expressed as the mean ± SD, *n* = 3.

## 4. Conclusions

The study aimed to discover new anti-inflammatory drugs from the roots of *M. malabathricum subsp.*
*normale* based on our previous works [18,37]. As expected, fourteen compounds were obtained from the tile plant for the first time, and compounds **1** and **11** are new compounds. In addition, this is the first report of compounds **1**–**4**, **8**, and **10**–**14** from the genus *Melastoma*, and compounds **1**, **2**, and **11**–**14** from the family Melastomataceae. Compounds **1**, **11**, and **12** showed anti-inflammatory activities, which make them potential anti-inflammatory drugs. The study of structure–activity relationship is helpful to quickly find out new anti-inflammatory drugs. The successful isolation and structure identification of compounds **1**–**14** not only provide materials for this experiment, but also contribute to the chemotaxonomic studies of the species of the genus *Melastoma*.

## Figures and Tables

**Figure 1 molecules-27-01521-f001:**
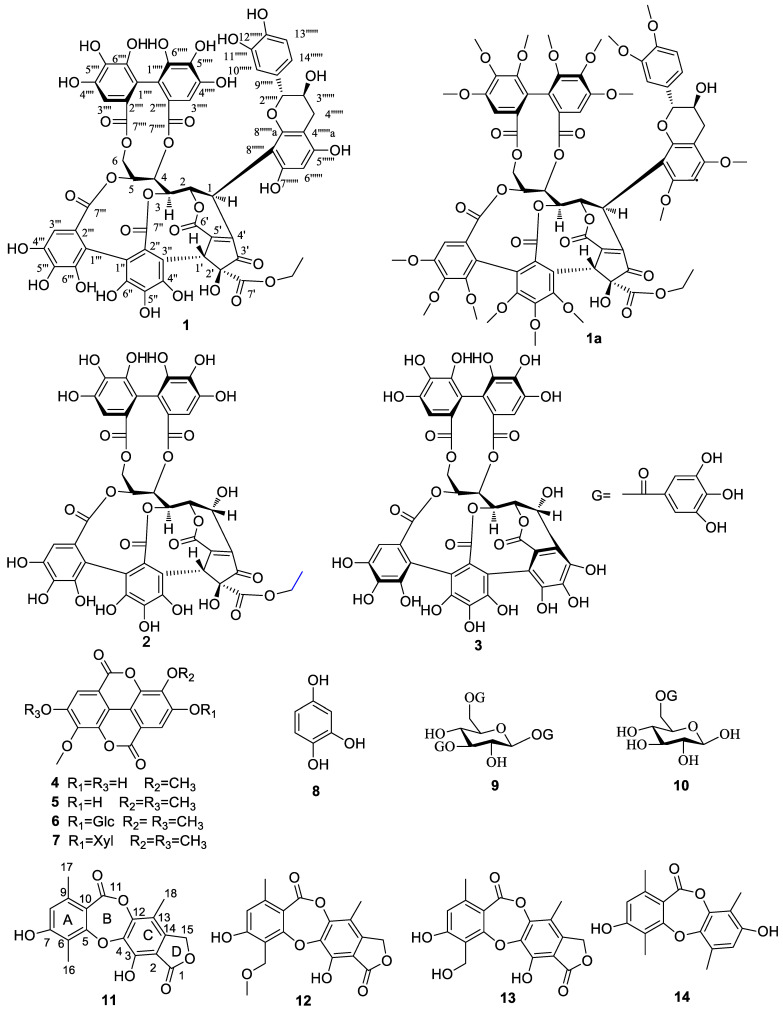
Structures of compounds **1**–**1****4**.

**Figure 2 molecules-27-01521-f002:**
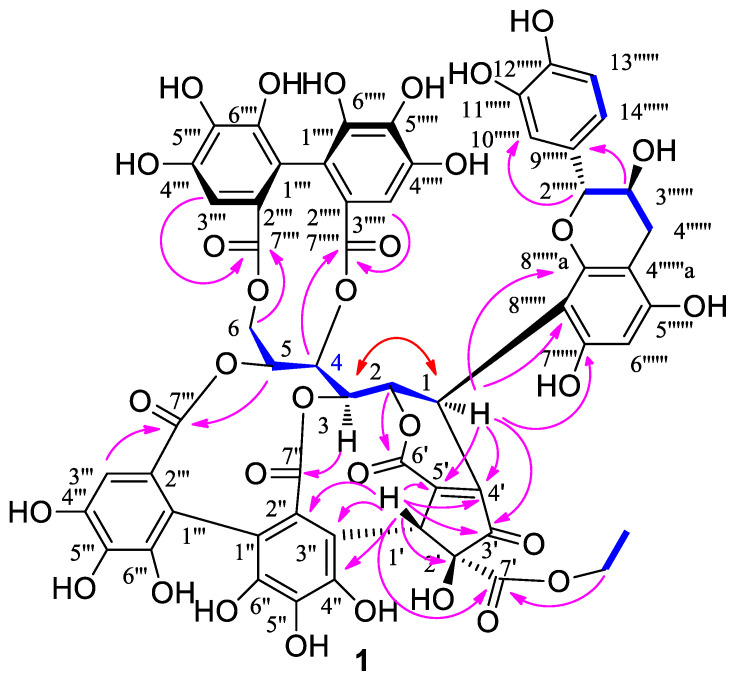
Key HMBC (arrows), ^1^H-^1^H COSY (bonds), and NOE (double arrows) correlations of **1**.

**Figure 3 molecules-27-01521-f003:**
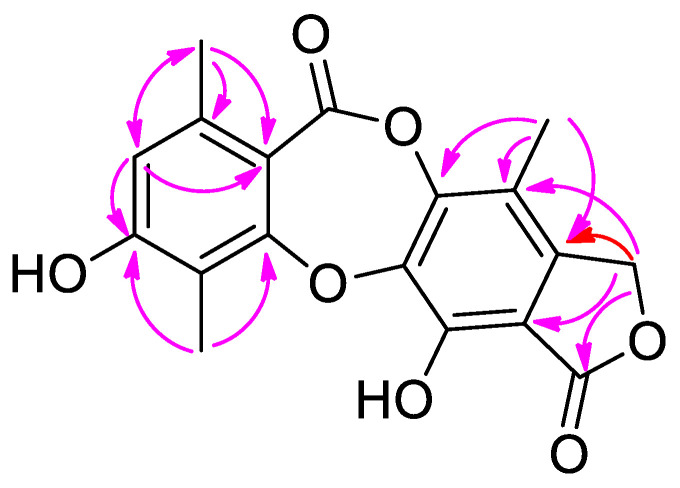
Key HMBC (arrows) of **11**.

**Table 1 molecules-27-01521-t001:** ^1^H (500 MHz) and ^13^C NMR (125 MHz) spectroscopic data for **1** in acetone-*d*_6_.

Pos.	δ_H_	δ_C_	Pos.	δ_H_	δ_C_	Pos.	δ_H_	δ_C_
Glc-1	4.36 s	33.9	HHDP-1′′′		114.1	Catechin-2′′′′′′	4.72 br s	82.4
2	4.96 s	83.2	2′′′		125.9	3′′′′′′	4.11 m	67.5
3	5.29 s	74.4	3′′′	6.80 s	108.9	4′′′′′′	2.83 d (16.1)	28.8
4	5.53 d (6.7)	68.7	4′′′		144.3		2.57 dd (16.1, 7.7)	
5	5.35 s	71.7	5′′′		137.2	4′′′′′′a		100.5
6a	4.81 dd (11.8, 6.2)	63.6	6′′′		145.2	5′′′′′′		154.0
6b	3.86 m		7′′′		167.8	6′′′′′′	6.00 s	96.4
Cp-1′	5.60 s	45.1	HHDP-1″″		115.1	7′′′′′′		154.9
2′		84.1	2″″		125.8	8′′′′′′		102.9
3′		201.4	3″″	7.02 s	108.8	8′′′′′′a		156.2
4′		146.8	4″″		144.8	9′′′′′′		131.3
5′		155.9	5″″		136.7	10′′′′′′	6.94 d (0.9)	114.2
6′		164.2	6″″		146.0	11′′′′′′		145.6
7′		170.3	7″″		168.2	12′′′′′′		145.1
HHDP-1″		115.8	HHDP-1′′′″		114.2	13′′′′′′	6.86 dd (8.2, 0.9)	115.9
2″		124.7	2′′′″		125.0	14′′′′′′	6.78 d (8.2)	120.6
3″		112.8	3′′′″	6.63 s	108.5	OCH_2_	4.21q (7.1)	62.9
4″		144.2	4′′′″		144.1	CH_3_	1.20 t (7.1)	14.3
5″		136.5	5′′′″		135.4			
6″		145.5	6′′′″		146.0			
7″		168.0	7′′′″		167.4			

**Table 2 molecules-27-01521-t002:** ^1^H (500 MHz) and ^13^C NMR (125 MHz) spectroscopic data for **11** in DMSO-*d*_6_.

Pos.	δ_H_	δ_C_	Pos.	δ_H_	δ_C_	Pos.	δ_H_	δ_C_
1		168.9	7		160.4	13		113.9
2		109.5	8	6.63, s	114.9	14		143.7
3		148.6	9		140.7	15	5.25, s	68.3
4		140.1	10		111.4	16	2.30, s	8.7
5		160.5	11		161.7	17	2.30, s	20.4
6		114.9	12		148.6	18	2.12, s	11.0

**Table 3 molecules-27-01521-t003:** The anti-inflammatory activities of compounds **1**–**12**.

Compound	IC_50_ (μM) ^a^
**1**	8.02 ± 0.35
**2**	>50
**3**	>50
**4**	>50
**5**	>50
**6**	>50
**7**	>50
**8**	21.32 ± 1.05
**9**	>50
**10**	>50
**11**	6.46 ± 0.23
**12**	9.82 ± 0.43
${\left[\alpha \right]}_{D}^{25}$ Dexamethasone	2.52 ± 0.26

^a^ Values present mean ± SD of triplicate experiments.

## Data Availability

Not applicable.

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
