# Peer review of "Structural Characterization and Assessment of Anti-Inflammatory Activities of Polyphenols and Depsidone Derivatives from Melastoma malabathricum subsp. normale"

_molecules, 2022, doi:10.3390/molecules27051521_

Round 1

Reviewer 1 Report

This manuscript is on the isolation of bioactivities of certain flavonoids and other compounds from Melastoma normale D. Don shrub present in several provinces of China. The authors have performed several characterization of the compounds. I have the following comments and suggestions:

Abstract: “investigated their”  should be “investigated for their”

There are incomplete sentences and typos in the whole manuscript that need must be corrected.

In table 3, only compound 1, 11, and 12 data are shown while in the table description it was written as compound 1-11. Kindly correct this.

Author Response

  1. Abstract: “investigated their”  should be “investigated for their”

--Answer: You are right. We have amended it.

  1. There are incomplete sentences and typos in the whole manuscript that need must be corrected.

--Answer: Thanks for your comment. We try our best to correct them.

  1. In table 3, only compound 1, 11, and 12 data are shown while in the table description it was written as compound 1-11. Kindly correct this.

--Answer: Thanks for your comment. We have corrected them.

Reviewer 2 Report

Dear Author,
Your work is very impressive. However, I have some suggestions and corrections.
1. Melastoma normale is a synonym of Melastoma malabathricum subsp. normale (D. Don) Karst.Mey(Ref: Plant of the world website). Is it possible to use the accepted name instead?
2. The sentence in line 63-65 ",indicating their particularity in this plant" might make the readers hard to understand.
3. Could you please explain more for the sentence line 99-100 in order that the reader would not have to find the explanation in Ref [31]?
4. Could you please label Cp-7' position in the figure 1?
5. Please correct the spelling of stenaphyllanin B in line 126.
6. The chemical shift of 13C-NMR of C-12 (139.5) (line 167) was not consistent with that of Table 2.

7. Line 184, ",indicating that CH3O- at C-18..." should be corrected to ",indicating that CH3O- at C-16...."

8. Line 256, is the "nondecamethyl ether" correct? Because it was 16 methoxy groups in 1a.
9. How did you perform gradient elution in Sephadex CC? Because the pore size of resin was changed along the gradient. In addition, the Sephadex resin should be settled.

10. Could you add more discussion on chemotaxonomic approach in genus Melastoma and the family Malastomataceae in aspect of complex tannins and depsidones.

Author Response

Dear Author,
Your work is very impressive. However, I have some suggestions and corrections.
1. Melastoma normale is a synonym of Melastoma malabathricum subsp. normale (D. Don) Karst.Mey(Ref: Plant of the world website). Is it possible to use the accepted name instead?

--Answer: Thanks for your comment. “Melastoma normale” was revised as “ Melastoma malabathricum subsp. normale (D. Don) Karst.Mey
2. The sentence in line 63-65 ",indicating their particularity in this plant" might make the readers hard to understand.

--Answer: You are right. And the sentence “indicating their particularity in this plant” was deleted.
3. Could you please explain more for the sentence line 99-100 in order that the reader would not have to find the explanation in Ref [31]?

--Answer:Thanks for your comment.”The large difference in chemical shifts between H-6a (δ 4.81) and 6b (δ 3.86) suggested one of the HHDP moieties was located at C-4 and C-6,” was revised as “The large difference in chemical shifts between H-6a (δ 4.81) and 6b (δ 3.86) suggested one of the HHDP moieties was located at C-4 and C-6, which can be explained by the anisotropic effect of a C-6 ester carbony| group. It is restrained to be rigidly coplanar with one of the C-6 methylene protons in the eleven-membered diester ring, so the proton was placed in a strongly deshielding environment.”
4. Could you please label Cp-7' position in the figure 1?

--Answer: You are very rigorous. We've labeled it.

5. Please correct the spelling of stenaphyllanin B in line 126.

--Answer: Thanks for your comment. “stenaphyllanin B” was revised as  “stenaphyllanine B”.

6. The chemical shift of 13C-NMR of C-12 (139.5) (line 167) was not consistent with that of Table 2.

--Answer: You are very serious. We've corrected it.

7.Line 184, ",indicating that CH3O- at C-18..." should be corrected to ",indicating that CH3O- at C-16...."

--Answer: Thanks for your reminding. We've corrected it.

8. Line 256, is the "nondecamethyl ether" correct? Because it was 16 methoxy groups in 1a.

--Answer: You are right. We've corrected it.

9. How did you perform gradient elution in Sephadex CC? Because the pore size of resin was changed along the gradient. In addition, the Sephadex resin should be settled.

--Answer: You are quite professional. We have corrected it according to your request.

10. Could you add more discussion on chemotaxonomic approach in genus Melastoma and the family Malastomataceae in aspect of complex tannins and depsidones.

--Answer: Thank you for your suggestion. We have added.

Reviewer 3 Report

In this manuscript, the authors purified the compounds in the roots of Melastoma normale and isolated and structurally characterized two new compounds and twelve known compounds. These compounds were tested for their in vitro anti-inflammatory activity inhibition of NO production in RAW264.7 cells.

The content of the manuscript is appropriate and sufficient to present the results of the study. However, the reviewer judged that the research described in the manuscript was not novel enough to be published in Articles of molecules.

A point that needs to be corrected:

The 13C NMR in supplementary compound 1 should be corrected because the phases are not sufficiently matched.

Author Response

A point that needs to be corrected:

The 13C NMR in supplementary compound 1 should be corrected because the phases are not sufficiently matched.

--Answer: Thanks for your comment. We've corrected it. It can be seen from the HSQC spectrum that deuterated solvent contains impurities.

Round 2

Reviewer 3 Report

I pointed out that the 13C NMR peak of compound 1 appears on the lower side, is this dept?

In any case, it is possible to check the chemical shift of the spectral peaks, which is fine.